# Bromodomain Inhibitor JQ1 Provides Novel Insights and Perspectives in Rhabdomyosarcoma Treatment

**DOI:** 10.3390/ijms23073581

**Published:** 2022-03-25

**Authors:** Irene Marchesi, Milena Fais, Francesco Paolo Fiorentino, Valentina Bordoni, Luca Sanna, Stefano Zoroddu, Luigi Bagella

**Affiliations:** 1Department of Biomedical Sciences, University of Sassari, Viale San Pietro 43/b, 07100 Sassari, Italy; iremarchesi@gmail.com (I.M.); faismilena@gmail.com (M.F.); fpfiorentino@gmail.com (F.P.F.); v.bordoni@studenti.uniss.it (V.B.); lusanna@uniss.it (L.S.); s.zoroddu3@studenti.uniss.it (S.Z.); 2Kitos Biotech Srls, Tramariglio, 07041 Alghero, Italy; 3Sbarro Institute for Cancer Research and Molecular Medicine, Centre for Biotechnology, College of Science and Technology, Temple University, Philadelphia, PA 19122, USA

**Keywords:** BET inhibition, rhabdomyosarcoma, MYC, BRD4, (+)-JQ1

## Abstract

Rhabdomyosarcoma (RMS) is the most common type of pediatric soft tissue sarcoma. It is classified into two main subtypes: embryonal (eRMS) and alveolar (aRMS). MYC family proteins are frequently highly expressed in RMS tumors, with the highest levels correlated with poor prognosis. A pharmacological approach to inhibit MYC in cancer cells is represented by Bromodomain and Extra-Terminal motif (BET) protein inhibitors. In this paper, we evaluated the effects of BET inhibitor (+)-JQ1 (JQ1) on the viability of aRMS and eRMS cells. Interestingly, we found that the drug sensitivity of RMS cell lines to JQ1 was directly proportional to the expression of MYC. JQ1 induces G1 arrest in cells with the highest steady-state levels of MYC, whereas apoptosis is associated with MYC downregulation. These findings suggest BET inhibition as an effective strategy for the treatment of RMS alone or in combination with other drugs.

## 1. Introduction

Rhabdomyosarcoma (RMS) is the most common form of pediatric soft tissue sarcoma, with an incidence of approximately 4.5 cases per 1 million children [1,2]. RMS cells are characterized by the expression of early myogenesis markers, but fail to complete the process with consequent incomplete differentiation and uncontrolled proliferation [3,4,5]. The 5-year survival rate of RMS patients is approximately 70% with a standard combination of chemotherapy, surgery, and radiotherapy [6]. The remaining 30% of patients die because of metastasis or tumor relapse with resistance to further cycles of therapy [2,7,8].

Several studies have shown that MYC and/or MYCN oncogenes are frequently amplified or overexpressed in RMS cells as well as in different types of sarcomas. Figure 1 shows MYC expression in several sarcoma cell lines (data obtained from Cancer Cell Line Encyclopedia (https://sites.broadinstitute.org/ccle/; accessed on 21 March 2022) and analyzed with R Studio software). These data support the hypothesis that suppression of MYC activity could be a therapeutic strategy for RMS tumors [9,10,11,12]. A pharmacological approach to suppress the expression or the activity of MYC and MYCN in cancer cells is represented by Bromodomain and Extra-Terminal domain (BET) protein inhibitors [13,14,15,16]. BET proteins are chromatin readers that recognize acetylated histones and interact with the transcription machinery, facilitating the transcription of target genes [17]. In malignant cells, BET proteins bind super-enhancer regions, driving the expression of oncogenes such as MYC [18]. The antiproliferative effects of BET inhibitors on RMS cells have been shown by two independent reports in preclinical models of alveolar RMS (aRMS), one of the two major RMS subtypes representing 23% of RMS cases [1,19,20]. In one of the two manuscripts, high sensitivity to BET inhibitor (+)-JQ1 (JQ1) in RH4 cells was reported, whereas reduced sensitivity was shown in RH30 cells; these are both PAX3-FOXO1 fusion-positive aRMS cells [19]. In the second report, it was shown that all tested PAX3-FOXO1 aRMS cell lines were highly sensitive to JQ1. The authors focused their attention on PAX3-FOXO1-mediated regulation of super enhancer-driven factors, among which are MYC and MYCN, to clarify the sensitivity of RMS cells to BET inhibition [20]. However, in the first report, RH30 cells were reported to be relatively resistant to JQ1 (IC50 > 10 µM), and this cell line was not assessed in the second one. Finally, sensitivity to BET inhibition in embryonal RMS, which account for 75% of RMS cases [1], has not yet been fully investigated. Therefore, we aimed to assess the effects of JQ1 in both aRMS and eRMS cells and evaluate the association between BET inhibition sensitivity and MYC levels.

## 2. Results

### 2.1. Expression of MYC Is Variable between RMS Cell Lines and Does Not Depend on the Fusion State of PAX3-FOXO1

In order to evaluate the expression of MYC proteins in RMS cells, Western blot on whole lysates from RD, A204, RH4, and SJCRH30 cells was performed (Figure 2). Whole lysate from small cell lung cancer H69 cells was added as a positive control for MYCN expression [21]. All tested RMS cell lines showed positive expression of MYC. RD and A204 embryonal RMS cells showed negative expression of MYCN, whereas a faint band was observed in alveolar RH4 and SJCRH30 RMS cells. Band intensities of MYC varied among RMS cell lines. RH4 showed the highest expression and RD the lowest one. No association between MYC levels and PAX3-FOXO1 fusion or RMS subtype were observed, because eRMS fusion-negative A204 cells showed similar MYC expression to aRMS fusion-positive SJCRH30 cells.

### 2.2. JQ1 Sensitivity Is Associated with MYC Steady-State Levels

To investigate the efficacy of BET inhibition on RMS cells, RD, A204, RH4 and SJCRH30 cells were treated with 11 concentrations of JQ1, ranging from 10 nM to 10 µM. Reduction of cell number and increase of cell death were observed in a dose dependent manner in all examined cell lines (Figure 3). GI50 was calculated to compare drug efficacy among distinct cell lines [22]. A value between 10 and 200 nM was observed in all tested cell lines after 72 h of treatment (Table 1). In RH4, A204, and SJCRH30 cells, similar GI50 values were also observed at 24 and 48 h of treatment, indicating that JQ1 reached its maximum effect in these cell lines within 24 h. In RD cells, a GI50 value around 3.5 µM and 400 nM was observed after 24 or 48 h of treatment, respectively, indicating maximum effects after a prolonged period of time. As shown in Figure 4, GI50 values were significantly associated with MYC protein levels assessed by Western blot, suggesting that high levels of MYC increased JQ1 vulnerability in these cell lines.

### 2.3. JQ1 Induce Cell Growth Arrest and Cell Death in RMS 3D Tumor Spheroids

In order to better predict the in vivo efficacy of JQ1 treatment on RMS cells compared to classic cell-based screens, cells were tested as 3D homotypic spheroid models of RMS [23]. Because RD formed viable spheroids whose volume did not increase over time, and SJCRH30 formed spheroids with reduced viability, the assay was conducted in homotypic spheroids originated from A204 and RH4 cells. Spheroids were treated with nine concentrations of JQ1 ranging from 20 nM to 5 µM. Reduction of spheroid size and increase of cell death were observed in a dose dependent manner in both examined models, with a prevalent cytostatic effect on A204 spheroids and a marked cytotoxic effect on RH4 spheroids (Figure 5).

Overall, the viability assay on tumor spheroids confirmed the results obtained in cells cultured as a 2D monolayer.

### 2.4. JQ1 Sensitivity Is Associated to Cell Cycle Arrest and Apoptosis

To assess the effects of JQ1 on the progression of cell cycle, the DNA content of RMS cells was assessed after 48 h of JQ1 treatment. Two treatment concentrations were chosen to exert similar phenotypes between cell lines, based on GI50 values (RH4, 50, and 100 nM; A204 and SJCRH30, 100, and 500 nM; RD, 500, and 2500 nM). In A204, RH4, and SJCRH30 cells, an increase of events in the G0/G1 phase was observed, indicating cell cycle arrest in the G1 phase (Figure 6A). In RD cells, an increase of events in subG1 phase was observed, but an arrest in a specific phase of cell cycle was not detected. An increase of events in subG1 phase was also observed in RH4 cells, but not in A204 or SJCRH30 cells.

Therefore, Western blot was performed to evaluate the markers of cell cycle arrest in the G1 phase (p21, p27), induction of apoptosis (cleaved PARP-1), and MYC levels. Levels of both p21 and p27 were increased in A204 and RH4 cells, whereas levels of p21 alone were increased in SJCRH30 cells, confirming the induction of G1 phase arrest in these cell lines (Figure 6B). Levels of the two G1 checkpoint regulators were not increased by JQ1 treatment in RD cells, confirming the absence of cell cycle arrest in this cell line. Cleavage of PARP-1 was detected in RD, RH4, and SJCRH30, indicating induction of apoptosis by JQ1 treatment (Figure 6B). Concomitant downregulation of MYC was detected in RD, RH4, and SJCRH30, but not in A204 (Figure 6B and Table 2). Overall, these results indicate that in cell lines characterized by high steady-state levels of MYC (RH4, SJCRH30, and A204), JQ1 treatment induces cell cycle arrest, while apoptosis is triggered in cells where concomitant downregulation of MYC is detected: RH4, RD, and SJCRH30 cells.

## 3. Discussions

Epigenetic abnormalities are involved in every aspect of tumor biology: development, progression, recurrence, and drug sensitivity [24].

Various studies have shown that RMS is characterized by numerous epigenetic alterations [25,26] and focused on targeting the epigenetic processes as a way of treatment for this disease [27,28,29,30,31].

BET domains recognize acetylated lysine in histones as well as in other proteins [32,33]. BRD4, one of the most studied member of the BET-containing protein family, is an epigenetic factor that is associated with active promoters and enhancers of transcriptionally active genes [18,34]. BRD4 is a functional bridge between acetylated chromatin and active transcription. It binds to acetylated chromatin through the bromodomain and recruits CyclinT/CDK9 complex and other mediators to promote gene expression [35]. BRD4 occupancy can be particularly elevated in highly-acetylated chromatin elements, termed super-enhancer [13,18]. In multiple myeloma and diffuse large B cell lymphoma, where high expression of MYC is a consequence of an active super-enhancer element proximal to MYC gene, BET inhibition induces strong MYC downregulation [18,36]. Because the acetylation of super-enhancers is cell type-specific, it is possible to consider that the sensitivity to BET inhibition differs according to their configuration [13].

A number of different strategies have been used to suppress the activity of MYC in cancer cells, some of which have progressed to clinical trials as BET inhibitors and G-quadruplex stabilizers (Quarflaxin). However, these inhibitors have shown poor bioavailability, and therapeutic selectivity remains a major concern for scientific research [37]. In fact, many compounds have shown high toxicity and suboptimal safety profiles. In addition, some of these compounds exhibited limited permeability and poor bioavailability due to their overly lipophilic or water-insoluble nature [38]. It is important to note that new therapeutic approaches are constantly being advanced based on the increasing understanding of the MYC gene.

The aim of this paper is to clarify the molecular mechanism of BET inhibition in RMS cells. In this paper, we show that (I) RMS cells that expressed relative high levels of MYC (RH4, A204, SJCRH30), reached the maximum cytotoxic effect within 24 h of treatment, whereas RD cells that expressed relative low levels of MYC reached the maximum effect after 72 h (Figure 3), and (II) the efficacy of BET inhibitor JQ1 in RMS cells was directly proportional to the levels of MYC protein (Figure 4). These results indicate that sensitivity to JQ1 in RMS cells is associated with MYC expression, regardless of alveolar or embryonal origin. A cell viability assay conducted in 3D homotypic RMS spheroids confirmed the results obtained in 2D culture (Figure 5).

We also demonstrated that BET inhibition by JQ1 induced G1 arrest in RMS cells with higher levels of MYC, whereas no effects on cell cycle has been detected in RD cells (Figure 6A). This result was confirmed by Western blot, which showed increased levels of p21 and/or p27 in JQ1-treated cells, with the exception of RD cells (Figure 6B).

Despite a low GI50 value and the strong G1 arrest observed, A204 failed to go into apoptosis after treatment with JQ1. Indeed, no cleaved PARP was detected in Western blot analysis. Moreover, MYC protein levels are not affected by JQ1 treatment (Figure 6B and Table 2). These data may suggest that intrinsic resistance mechanisms in A204 can prevent MYC downregulation and apoptosis. In order to identify the pathways involved in the lack of response to BET inhibitors, this hypothesis should be confirmed in other cellular models and through in vivo experiments. However, several mechanism of resistance to BET inhibition have already been discovered in distinct tumors [33]. For instance, a mechanism involving Wnt/β-catenin, Hedgehog, MAPK signaling or TGF-β pathway compensates for the effects of BRD4 inhibition on MYC expression or protein stability in leukemia, hepatocellular carcinoma, and pancreatic and colorectal cancer [39,40,41,42,43,44].

Furthermore, apoptosis resistance can be induced by alternative phenotypic response as autophagy [45], a mechanism that protect from mitochondrial damage, a key event for BET inhibition [46] or kinome reprogramming [41].

Other studies are needed to identify the resistance mechanisms able to allow the evaluation of the combinations of inhibitors to maximize BET inhibition efficacy.

## 4. Materials and Methods

### 4.1. Cell Culture

RD, SJCRH30, and H69 were purchased from American Type Culture Collection (ATCC, Rockville, MD, USA), RH4 were kindly provided by Dr. S.V. Forcales, and A204 was purchased from CLS Cell Lines Service GmbH (Eppelheim, Germany). RD and A204 were cultured in DMEM supplemented with 10% FBS, 1% L-glutamine, and 1% antibiotics–antimycotic solution. SJCRH30, RH4, and H69 were cultured in RPMI and supplemented with 10% FBS, 1% L-glutamine, and 1% antibiotics–antimycotic solution; cells were cultured at 37 °C in humidified atmosphere, 5% CO_2_.

### 4.2. Kinetics of Cytotoxicity and Cell Proliferation

Cell Proliferation and cytotoxicity assay was performed as described in [22]. Five hundred cells suspended in 20 μL of phenol-red free medium supplemented with SiR-Hoechst 0.5 μM (Spirochrome, Stein am Rhein, Switzerland, CH) [47], and CellTox^TM^ Green Dye 1x (Promega, Madison, WI, USA), were seeded in a 384-well flat, clear bottomed, black microplate (Corning Inc., Corning, NY, USA). After 18–24 h, cells were treated with 10 μL of fresh culture medium containing JQ1 (Tocris Bioscience, Bristol, UK) in the range of 10–0.01 µM. Cell plating, serial dilutions of JQ1, and treatment of the cells were performed using an automated liquid handling platform (Gilson Pipetmax^®^, Middleton, WI, USA).

Live-cell imaging in far-red fluorescence, green fluorescence, and phase contrast were carried out at 37 °C, 5% CO_2_ using an automated digital widefield microscope and an associated gas controller (BioTek Cytation 5, Winooski, VT, USA). Pictures were collected immediately before the beginning of treatment and after 24, 48, and 72 h using 4× objective. For each well, four images were acquired to cover the entire well area. Image assembly and cell count were performed using BioTek Gen5 software.

The number of total cells was calculated by counting nuclei stained in far-red; the number of dead cells was calculated by counting nuclei stained in green. The number of live cells has been determined by subtracting the number of dead cells from the number of total cells.

The GI50 value, representing the drug concentration required to reduce cell proliferation by 50%, was calculated with R software, loess/approx functions [48].

### 4.3. Kinetics of 3D Tumor Spheroid Growth and Survival

Five hundred A204 and RH4 cells were seeded into 384-multi-well ultra-low attachment plates in a volume of 20 μL per well, allowing the formation of a single spheroid in each well (Corning Inc., Corning, NY, USA). Tested cell lines formed tight, round shaped, spheroids after 72 h of incubation at 37 °C, 5% CO_2_. RMS spheroids were treated with nine dilutions of JQ1, in the range of 5–0.02 µM, and CellTox 0.33x for cell death quantification, in a final volume of 30 μL. Immediately after the beginning of treatment and every 24 h for 72 h, live-cell images in a brightfield and GFP filter (led cube 465 nm, filter cube excitation 469 ± 25, emission 525 ± 25) were collected using BioTek Cytation 5, 4× objective (BioTek Intruments, Winooski, VT, USA). For each spheroid, three images at 50 μM-distance focal planes were acquired and merged. Spheroid area and green fluorescence intensity (GFI), limited to spheroid area, were calculated using Gen5 software. Spheroid volumes (V) were calculated using the formula 4/3*Area*RADQ(Area/π) [49]. Relative volumes were calculated by normalizing volumes at each time point to the volume of the same spheroid immediately after the beginning of treatment (V/V_0_). Spheroid death was calculated as (GFI-GFI_0_)/Area, where GFI was identified as the green fluorescence intensity at each time point and GFI_0_ is the GFI immediately before the beginning of treatment. At least three spheroids were treated for each condition. Results show mean value +/− standard error. The RD50 value, representing the drug concentration required to reduce population doubling time by 50%, was calculated with R software, loess/approx functions [22].

### 4.4. Cell Cycle Analysis

Cell cycle analysis was performed as described in [50]. Briefly, cells were washed in PBS, fixed with 70% ice-cold ethanol, and incubated at −20 °C overnight. Fixed cells were washed in PBS and stained with a solution of propidium iodide 5 μg/mL (Sigma-Aldrich, Saint Louis, MO) and RNAse 20 μg/mL (Sigma-Aldrich, Saint Louis, MO, USA) in PBS. Samples were incubated overnight at 4 °C and DNA content of at least 20,000 cells was evaluated with a BD AccuryC6 Flow Cytometer (Beckton Dickinson, Franklin Lake, NJ, USA). Data were analyzed with R-BiocManager software (flowCore/flowViz package).

### 4.5. Western Blot

Cells were lysed using a solution of TRIS-HCl pH 8 20 mM, NaCl 137 mM, glycerol 10% Nonidet P-40 1%, EDTA 2 mM, and Protease Inhibitor Cocktails. Protein concentration was determined by Bradford assay (Biorad), using BSA as a standard.

An amount of 30 µg of proteins was resolved in 8% or 15% SDS acrylamide gels and transferred to a nitrocellulose membrane. After one hour in a blocking solution (non-fat dry milk 5% in TBS 0.1% Tween 20), membranes were incubated overnight at 4 °C with a solution of BSA in TBS 0.1% Tween 20 and the following primary antibodies: anti-p21, anti-Parp purchased from Cell Signaling (Boston, MA, USA), anti-p27, anti-MYC and anti-MYCN purchased from Santa Cruz Biotechnology (Santa Cruz, CA, USA), and anti-α tubulin purchased from Calbiochem (San Diego, CA, USA). Membranes were incubated with secondary peroxidase-conjugated antibodies (Pierce, Thermofisher Scientific, Waltham, MA, USA), and the signal was detected using ECL Western Blotting Substrate (Pierce Thermofisher Scientific, Waltham, MA, USA). Protein levels were evaluated with ImageJ software.

### 4.6. Datasets

The following datasets were used for gene expression analyses in different sarcomas: The cancer cell line encyclopedia (CCLE) project dataset is a compilation of gene expression data from human cancer cell lines [51]. The CCLE subset for Sarcomas contains MYC gene expression data from different cell lines, obtained from the CCLE portal (https://sites.broadinstitute.org/ccle/; accessed on 21 March 2022).

## Figures and Tables

**Figure 1 ijms-23-03581-f001:**
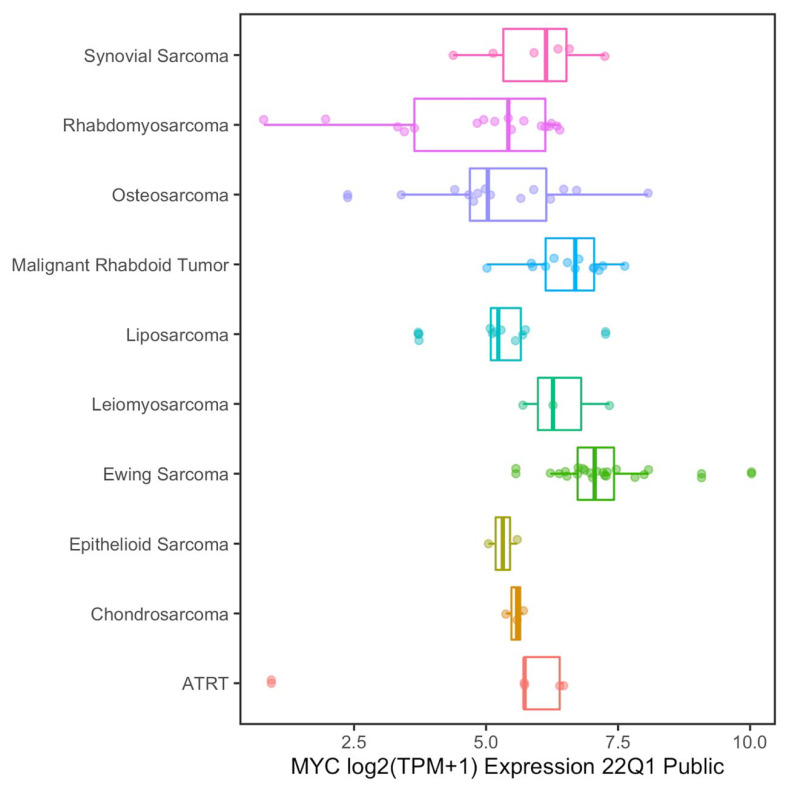
MYC expression in Cancer Cell Line Encyclopedia cell lines. BoxPlot displaying RNA-seq expression levels from CCLE Cancer Samples. TPM: Transcripts Per Kilobase Million.

**Figure 2 ijms-23-03581-f002:**
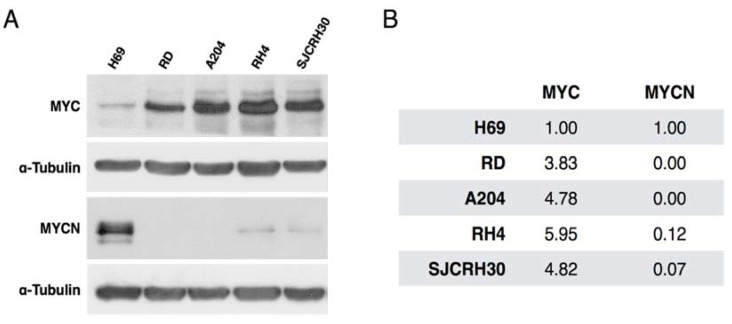
Expression of MYC and MYCN in RMS cell lies. (**A**) Western blot analysis performed in 4 RMS cell lines using antibodies for MYC and MYCN. H69 protein extract was used as positive control for MYCN. α-tubulin was used as housekeeping gene. (**B**) Protein quantification obtained with ImageJ software. Protein levels of MYC and MYCN for each sample were normalized with α-tubulin protein expression and results are expressed relative to H69 MYC or MYCN expression.

**Figure 3 ijms-23-03581-f003:**
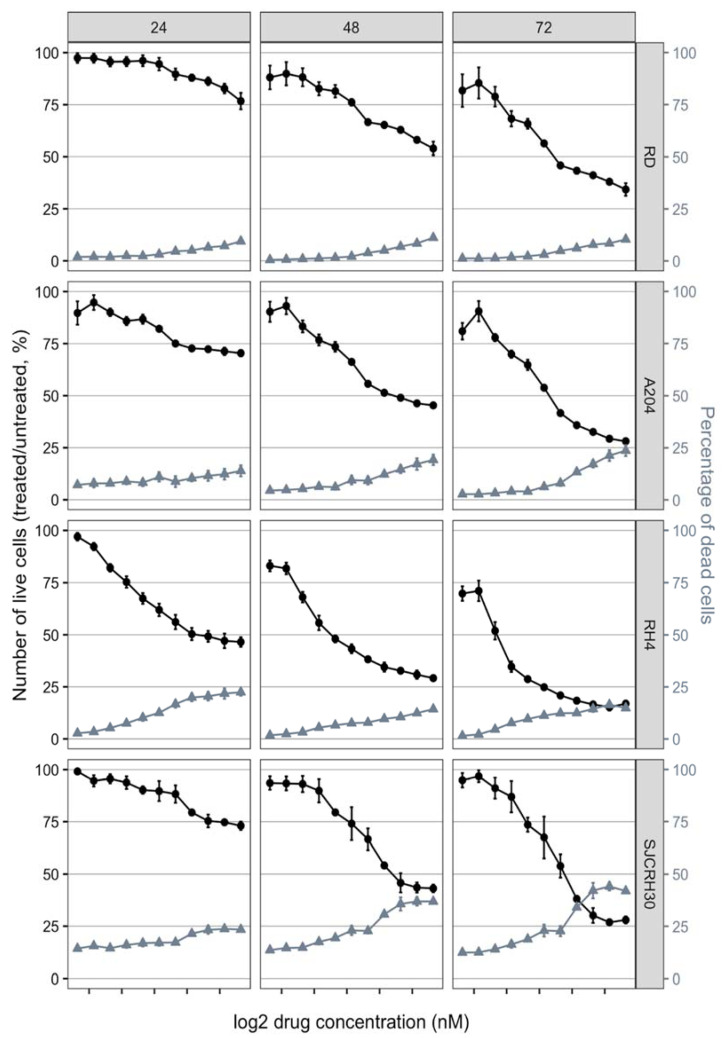
JQ1 reduce cell viability of RMS cell lines in a dose dependent manner. Images of cells at 24, 48, and 72 h of treatment were acquired, and multiple counting of living and dead cells was performed for each treatment, as reported in the Materials and Methods section. Cell viability, indicated with filled circles, was calculated as relative cell number (number of live cells in treated samples/number of live cells in untreated samples × 100). Percentage of death, indicated with grey filled triangles, was calculated as number of dead cells/total cells × 100.

**Figure 4 ijms-23-03581-f004:**
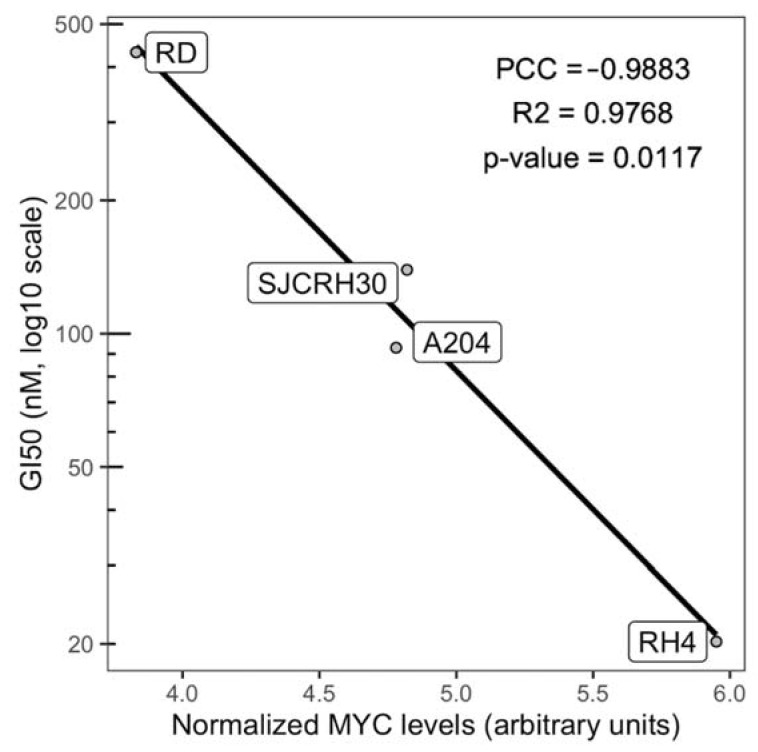
JQ1 sensitivity to JQ1 is related with MYC expression. Scatter chart of MYC levels and JQ1-induced GI50 in RMS cell lines after 48 h of treatment. Pearson’s Correlation Coefficient (PCC), R-square, and *p*-value were calculated using R software, function cor.test/lm.

**Figure 5 ijms-23-03581-f005:**
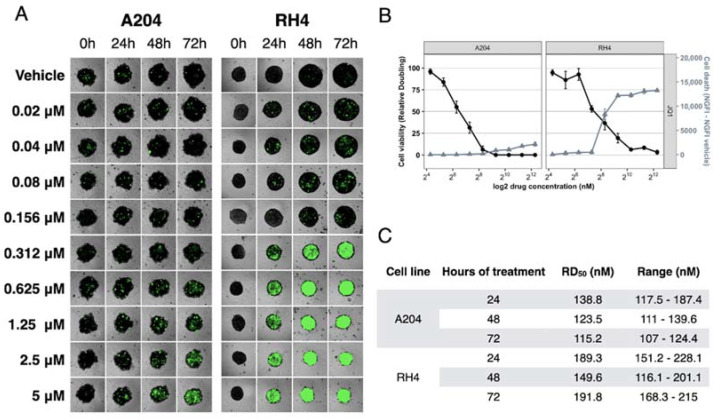
JQ1 reduce cell viability of RMS spheroids. Analysis of viability of 3D tumor spheroids originated from A204 and RH4 cell lines. (**A**) Image of a representative spheroid for each cell line, drug concentration, and time point. (**B**) Areas and GFP emissions (dead cells) were acquired for each spheroid at 72 h of treatment. Cell Viability is indicated with filled circle, and cell death is indicated with grey filled triangles. (**C**) RD50 calculated after 24, 48, and 72 h of treatment.

**Figure 6 ijms-23-03581-f006:**
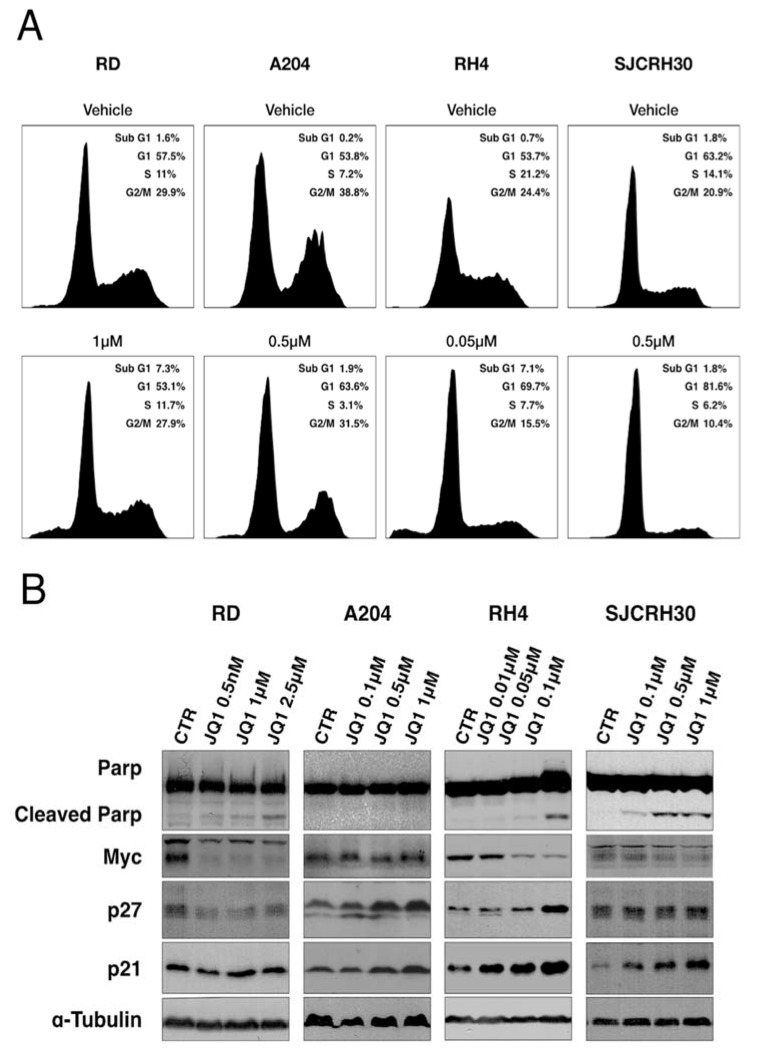
JQ1 induced G1 arrest and apoptosis in RMS cell lines. (**A**) Cell cycle profiling of RMS cells treated with JQ1. (**B**) Western blot analysis performed in RMS cell lines treated with different concentration of JQ1, using antibodies for MYC, PARP, p21, and p27. α-tubulin was used as the housekeeping gene.

**Table 1 ijms-23-03581-t001:** GI50 values after 24, 48, and 72 h of JQ1 treatment. Ranges in parentheses indicate mean ± se.

Cell Line	Hours of Treatment		
	**24**	**48**	**72**
**RD**	3486.5 (2200.4–6405.9)	431.7 (361.1–511.5)	103.7 (77.2–134.8)
**A204**	23.1 (31.5–111.1)	93.0 (73.6–113.9)	88.4 (45.6–102.3)
**RH4**	27.5 (24.9–30.4)	20.2 (17.5–23.1)	11.7 (<10–14.2)
**SJCRH30**	137.8 (52.2–303.7)	139.4 (102.8–191.5)	198.2 (146.7–256)

**Table 2 ijms-23-03581-t002:** Densitometric analysis of protein levels of Parp, Cleaved parp, MYC p27, and P21. The samples were normalized with α-tubulin protein expression, and results are expressed as relative to untreated cells. Protein quantification obtained with ImageJ software.

Cell Line	JQ1 Concentration (nM)	PARP	CLEAVED PARP	MYC	P27	P21
**RD**	0	1	1	1	1	1
	0.5	1.08	1.48	0.31	0.60	1.03
	1	0.62	0.77	0.19	0.44	0.90
	2.5	0.86	1.73	0.23	0.64	1.59
**A204**	0	1	0	1	1	1
	0.5	2.08	0.00	2.13	2.59	1.45
	1	0.89	0.00	0.93	2.34	1.19
	2.5	1.23	0.00	1.41	2.03	1.31
**RH4**	0	1	1	1	1	1
	0.5	0.81	0.72	0.93	1.05	1.97
	1	1.17	0.96	0.32	1.14	2.52
	2.5	1.77	3.02	0.13	1.72	1.93
**SJCRH30**	0	1	1	1	1	1
	0.5	0.74	1.52	0.71	0.87	1.84
	1	0.63	4.22	0.56	0.85	2.94
	2.5	0.61	3.87	0.32	1.06	3.76

## Data Availability

Not applicable.

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
