# Peer review of "Bromodomain Inhibitor JQ1 Provides Novel Insights and Perspectives in Rhabdomyosarcoma Treatment"

_ijms, 2022, doi:10.3390/ijms23073581_

Round 1

Reviewer 1 Report

Authors described efficacy MYC (BET) inhibitor in RMS cells. In this study, to clarify the efficacy of the inhibiter, they conducted functional analyses including cell viabilities and cell cycle profiling.  I thought this article provides novel clinical and research information for both the treatment of RMS patients and MYC (BET) inhibitor. However, there are some concerns.

Major)

・Authors should add information regarding MYC  expressions in the other types of sarcomas. I recommend that authors should analyze and confirm the MYC expressions in huge data-base including Cancer Cell Line Encyclopedia (CCLE) and The Cancer Genome Atlas (TCGA).

・It have been known that MYC inhibitors might cause serious side effects by inhibiting the growth of normal tissues, and that the design of MYC inhibitors have been practically difficult. Therefore, to avoid reader's misunderstanding, authors should add information regarding both MYC expressions in normal tissues and these concerns. Also, authors should conduct in-vivo assays, because they would achieve the development of clinical applications as therapeutic targets.

Author Response

Detailed Response to Reviewers:

Reviewer #1 (Reviewer Comments to the Author):

Authors described efficacy MYC (BET) inhibitor in RMS cells. In this study, to clarify the efficacy of the inhibiter, they conducted functional analyses including cell viabilities and cell cycle profiling.  I thought this article provides novel clinical and research information for both the treatment of RMS patients and MYC (BET) inhibitor. However, there are some concerns.

Major:

  • Authors should add information regarding MYC expressions in the other types of sarcomas. I recommend that authors should analyze and confirm the MYC expressions in huge data-base including Cancer Cell Line Encyclopedia (CCLE) and The Cancer Genome Atlas (TCGA).

Response to Reviewer #1 (Point 1): We thank the referee for this consideration and agree on the correction proposed. We performed analyses using the Cancer Cell Line Encyclopedia (CCLE) database as suggested observing interesting results. The figure has been included within the manuscript (57-60). Moreover, we have modified the “Introduction” (33-36) and “Materials and methods” (278-282) sections accordingly.

  • It have been known that MYC inhibitors might cause serious side effects by inhibiting the growth of normal tissues, and that the design of MYC inhibitors have been practically difficult. Therefore, to avoid reader's misunderstanding, authors should add information regarding both MYC expressions in normal tissues and these concerns. Also, authors should conduct in-vivo assays, because they would achieve the development of clinical applications as therapeutic targets.

Response to Reviewer #1 (Point 2): We thank referee for this accurate observation. In the “Discussion” section, we have added information regarding the difficulties of clinical application of MYC inhibitors (170-178). Furthermore, it is our interest to complete the understanding of these mechanisms through in vivo assays. However, we are planning to perform this part of the analysis and include it in a future work in order to add important clinical considerations in the therapeutic field.

Reviewer 2 Report

Dear authors

the topic is relevant and interesting. The results of this work provide a deeper insight as Bromodomain and Extra-Terminal motif inhibition can be an effective strategy for the treatment of Rhabdomyosarcoma alone or in combination with other drugs.
The article is well written and the text is clear and easy to read. Discussion is consistent with the evidence and arguments presented.

But is required:

point 1
insert the paragraph of the conclusions as required by the IJMS format

point 2
insert the densitometric analysis also for the western blots of figure 5

Author Response

Detailed Response to Reviewers:

Reviewer #2 (Reviewer Comments to the Author): Dear authors, the topic is relevant and interesting. The results of this work provide a deeper insight as Bromodomain and Extra-Terminal motif inhibition can be an effective strategy for the treatment of Rhabdomyosarcoma alone or in combination with other drugs. The article is well written and the text is clear and easy to read. Discussion is consistent with the evidence and arguments presented.

But is required:

1. Insert the paragraph of the conclusions as required by the IJMS format

Response to Reviewer #2 (Point 1): We thank the Referee for this important consideration. However, on the "Instructions for Authors" page indicated by the IJMS, the "conclusion" section does not need to be in the manuscript, but we have added a required concluding sentence in the discussion section (204-205).

2. Insert the densitometric analysis also for the western blots of figure 5

Response to Reviewer #2 (Point 2): We thank the referee for this consideration and agree on the correction proposed. We performed the densiometric analysis and added a table which can be found in the text (150-153).

Round 2

Reviewer 1 Report

Authors have addressed my all concerns.